# Widespread bacterial lysine degradation proceeding via glutarate and *L*-2-hydroxyglutarate

Sebastian Knorr[1,2], Malte Sinn[1,2], Dmitry Galetskiy[1], Rhys M. Williams[3], Changhao Wang[1], Nicolai Müller[3], Olga Mayans[2,3], David Schleheck[2,3] & Jörg S. Hartig[1,2]

Lysine degradation has remained elusive in many organisms including *Escherichia coli*. Here we report catabolism of lysine to succinate in *E. coli* involving glutarate and *L*-2-hydroxyglutarate as intermediates. We show that CsiD acts as an α-ketoglutarate-dependent dioxygenase catalysing hydroxylation of glutarate to *L*-2-hydroxyglutarate. CsiD is found widespread in bacteria. We present crystal structures of CsiD in complex with glutarate, succinate, and the inhibitor N-oxalyl-glycine, demonstrating strong discrimination between the structurally related ligands. We show that *L*-2-hydroxyglutarate is converted to α-ketoglutarate by LhgO acting as a membrane-bound, ubiquinone-linked dehydrogenase. Lysine enters the pathway via 5-aminovalerate by the promiscuous enzymes GabT and GabD. We demonstrate that repression of the pathway by CsiR is relieved upon glutarate binding. In conclusion, lysine degradation provides an important link in central metabolism. Our results imply the gut microbiome as a potential source of glutarate and *L*-2-hydroxyglutarate associated with human diseases such as cancer and organic acidurias.

[1] Department of Chemistry, University of Konstanz, Konstanz 78457, Germany. [2] Konstanz Research School Chemical Biology (KoRS-CB), Konstanz 78457, Germany. [3] Department of Biology, University of Konstanz, Konstanz 78457, Germany. These authors contributed equally: Sebastian Knorr, Malte Sinn. Correspondence and requests for materials should be addressed to J.S.H. (email: joerg.hartig@uni-konstanz.de)

For many organisms, lysine degradation has remained as a white spot on the metabolic map. In *E. coli*, a lysine decarboxylase activity has been described[1,2]. Further, degradation of cadaverine (Cad) to glutarate (GA) has been proposed in pseudomonads in 1977[3]. We initiated our study when being interested in the function of *Escherichia coli* protein CsiD (carbon starvation induced protein D). Its gene *csiD* is the first of a five-gene operon in *E. coli* (*csiD-lhgO-gabDTP*) (illustrated in Fig. 1), whose expression and regulation has been studied in great detail. The whole operon is specifically induced in stationary phase (carbon starvation) while the *gabDTP* genes are induced also separately in response to nitrogen starvation. Immediately downstream of the operon, *csiR* encodes a transcription factor that represses the *csiD* operon[4]. In addition to CsiR, the *csiD* operon is controlled by cAMP-CRP, leu-LRP and H-NS[5,6]. The CsiD protein belongs to the non-haem Fe(II)-dependent oxygenase family (protein family PF08943), but the native substrate(s) of the predicted enzyme and its role during stationary phase of *E. coli* remained unknown. A crystal structure of CsiD that was solved in a structural genomics effort suggested that CsiD functions as an α-ketoglutarate (αKG)-dependent dioxygenase[7]. Since the subsequent gene of the operon (*lhgO*, Fig. 1) has been described as an *L*−2-hydroxyglutarate (L2HG) oxidase[8], we hypothesized whether CsiD may produce L2HG by hydroxylation of GA, a compound that has so far been considered as a 'dead-end' metabolite[9]. Here we demonstrate that lysine is degraded via cadaverine to GA by a series of promiscuous aminotransferase and dehydrogenase reactions. GA is subsequently hydroxylated by CsiD and the product L2HG is oxidised to αKG by the dehydrogenase LhgO that couples to the respiratory chain by reducing the quinone pool. Furthermore, we show that repression of the CsiD operon by the transcription factor CsiR is selectively relieved by glutarate.

## Results

**Characterisation of glutarate hydroxylase CsiD.** We purified CsiD and demonstrated by NMR (Fig. 2a and Supplementary Figures 1, 2) and LC-MS that it indeed hydroxylates GA to 2-hydroxyglutarate, while the co-substrate αKG is converted to succinate (SA) (and $CO_2$), as is common for this enzyme class[10,11]. By derivatisation of the product with diacetyl-L-tartaric anhydride[12] we demonstrated that L2HG is produced in a highly stereospecific manner; no *D*-2-hydroxyglutarate was detectable (Supplementary Figure 3). This finding is in accordance with the reported specificity for the *L*-enantiomer of the subsequent LhgO-catalysed reaction[8]. When the reaction was measured with a Clark $O_2$ electrode, we determined a specific activity of $53 +/- 3$ µmol min$^{-1}$ mg$^{-1}$ and an apparent $K_m = 650 +/- 20$ µM for GA and $K_m = 100 +/- 7$ µM for αKG (Fig. 2b, c). Other dicarboxylic acids (oxalate, malonate, SA, adipate, and pimelate tested) were not converted. The physiological role of CsiD as glutarate-metabolising enzyme in the stationary phase of *E. coli* was confirmed when we tested its *csiD* knockout strain (*ΔcsiD*) by LC-MS of small-molecule extracts: with carbon starvation and entry into the stationary phase, the intracellular concentration of GA accumulated to much higher levels in the *ΔcsiD* strain than compared to the wildtype (Fig. 2d). While it is commonly encoded in *Enterobacteriaceae*, CsiD is found also in genomes of many other proteobacteria and bacilli (Supplementary Figure 4).

CsiD as characterised glutarate hydroxylase is also interesting with respect to the two structurally similar substrates GA and αKG. In addition, the product of the reaction, L2HG, is known as an oncometabolite inhibiting αKG-dependent dioxygenases such as TET-type and Jmjc-type demethylases[13,14]. We did observe weak inhibition of CsiD by its product L2HG (Supplementary

Figure 5) in contrast to the aforementioned representatives of the same enzyme class. In order to shed more light on this interesting finding, we solved the atomic structure of CsiD in complex with its substrate (GA), its product (SA) as well as in complex with the αKG-analog N-oxalylglycine (NOG) as inhibitor, by X-ray crystallography (Fig. 2e, Extended Fig. 6, Supplementary Table 1). The crystal form obtained (with symmetry $P42_12$) contains two molecular copies of CsiD in its asymmetric unit. These two non-crystallographic copies are identical (RMSD = 0.082 Å). The biological tetrameric form of CsiD is generated by the symmetry of the crystallographic lattice, as it was previously the case in structures lacking the by then unknown substrate[7,15]. The enzyme protomer adopts a distorted jelly-roll fold composed of a β-sheet core flanked by α-helices, as previously described. The iron ion is bound to the active site of CsiD by residues His160, Asp162, and His292 and three solvent molecules that complete an octahedral coordination sphere (Supplementary Figure 6a). In ligand-bound structures, one or more solvent molecules become replaced by interacting oxygen atoms from the ligands.

Ligand-bound CsiD structures revealed two binding sites located at opposite sides of the $Fe^{2+}$ ion (Fig. 2e and Supplementary Figure 6b-d). In the substrate site (here termed site I), GA is directly coordinated by the $Fe^{2+}$ ion via an oxygen atom from one of its terminal carboxyl groups. The other carboxyl group forms a salt bridge with a conserved residue Arg311 and the main chain nitrogen of Gly163 (Supplementary Figure 6b). The co-substrate site (here termed site II) is seen in crystal structures in this work occupied by (a) the non-processable αKG-analogue NOG (where atom C3 is substituted by nitrogen compared to αKG) or (b) the co-product SA. NOG binds the $Fe^{2+}$ ion via the oxo- and a terminal oxygen atom of its oxalyl moiety as well as forming a salt bridge with residue Arg309 (Supplementary Figure 6c). SA is bound to $Fe^{2+}$ via a terminal carboxyl group in a bidentate fashion (Supplementary Figure 6d) and further interacts with Arg309 through a low-occupancy solvent molecule that occupies the site that is generated by decarboxylation of αKG to SA, mimicking the interaction to the NOG oxygen in that complex. Conformational changes in the CsiD protein were not observed in any of the complexed structures, independently of whether ligand site I or site II was occupied. Importantly, in structures with site I or II occupied, always the cognate ligand for the respective site was observed with the other site unoccupied. This finding demonstrates a high degree of specificity of CsiD regarding the binding of the structurally similar substrate, co-substrate, and products.

**LhgO is a membrane bound quinone-dependent oxidoreductase.** We next considered the fate of L2HG in *E. coli*. The second protein of the operon, LhgO, has been described as a FAD-dependent *L*-2-hydroxyglutarate oxidase, which utilizes molecular oxygen to yield αKG and hydrogen peroxide[8]. However, we could not detect $O_2$ consumption with L2HG using the purified recombinant and active (see below) LhgO enzyme. LhgO is found very widespread with homologs in most eubacteria, archaea, and eukaryotes. The human (41% identity with *E. coli* LhgO over full amino acid sequence) and *A. thaliana* homologs are described as L2HG dehydrogenases[16,17] that localize to the inner mitochondrial membrane. Further, for the plant enzyme a coupling of the electron transfer to the respiratory chain has been proposed[17].

Hence, we tested whether the *E. coli* LhgO protein may be a dehydrogenase that feeds the high-energy electrons (reported $E^0 = 19 +/- 8$ mV)[8] from the oxidation of L2HG via ubiquinone to the respiratory electron transport chain (Fig. 1): Purified recombinant LhgO directly reduced ubiquinone (Supplementary

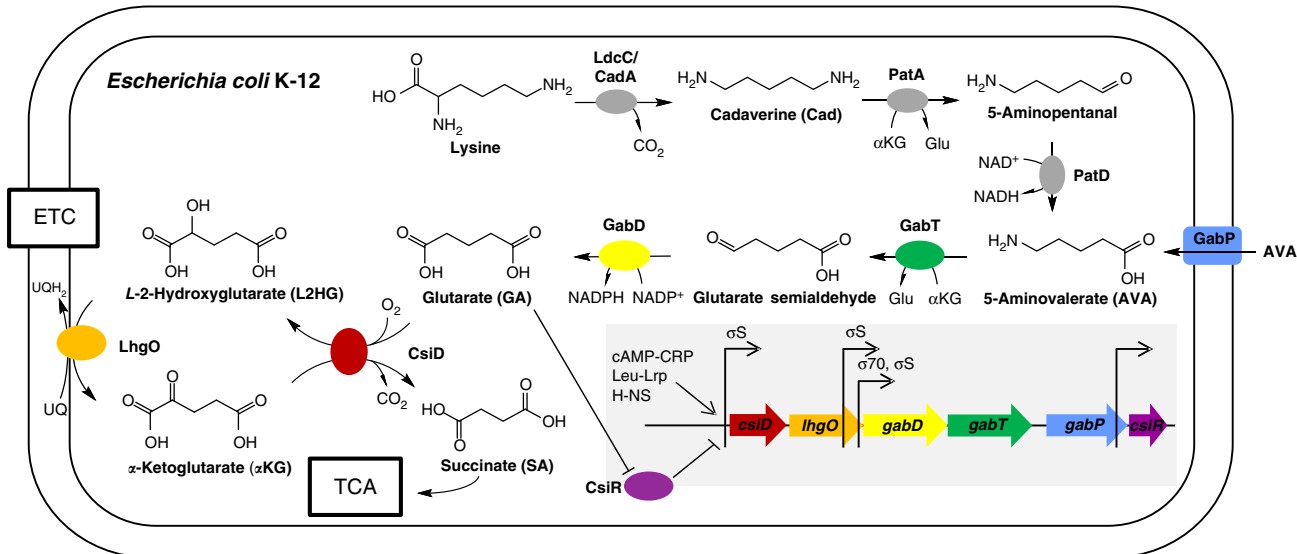

**Fig. 1** Catabolism of lysine to succinate in *Escherichia coli* K-12. Summary of the central metabolic pathway discovered in this study and of the corresponding gene cluster with its regulation (grey inset) in *E. coli*. We show that *E. coli* CsiD (carbon starvation induced protein D) functions as a α-ketoglutarate-dependent, $CO_2$- and succinate-forming glutarate hydroxylase, which produces *L*-2-hydroxyglutarate, and that *E. coli* LhgO (*L*-2-hydroxyglutarate oxidase) acts as a membrane-bound, ubiquinone-linked, α-ketoglutarate-producing *L*-2-hydroxyglutarate dehydrogenase. Effectively, glutarate is converted to succinate by these two reactions. We also show that *E. coli* feeds glutarate into these reactions from 5-aminovalerate by promiscuous GabT, when acting as 5-aminovalerate transaminase, and by promiscuous GabD, when acting as glutarate seminaldehyde dehydrogenase. Thus, these GabT-GabD-CsiD-LhgO reactions link the central metabolism (TCA cycle) with the yet incompletely known lysine catabolism in *E. coli*, that is, with LdcC and CadA as lysine decarboxylases, PatA (putrescine transaminase) acting as cadaverine transaminase and PatD as 5-aminopentanal dehydrogenase[1, 2, 26, 28]. The catabolism of lysine by this pathway is specifically induced in the stationary phase of *E. coli* (see text). The inbox shows the structure and regulation of the *csiD-lhgO-gabD-gabT* operon, which include also genes *gabP* encoding for a gamma-aminobutyrate (GABA) transporter that also imports the C5-homologue 5-aminovalerate[55] and *csiR* encoding for a ligand-dependent transcription factor. Arrows represent transcription start sites (TSS) of different transcripts. Transcription of the operon is known to be enhanced by regulators cAMP-CRP, leu-LRP and H-NS[5, 6], and repressed by CsiR[4]. We demonstrated that this repression is relieved specifically upon binding of glutarate to CsiR

Figure 7) in a L2HG-dependent manner with a specific activity of $0.33+/-0.002\ \mu mol\ min^{-1}\ mg^{-1}$. Enzyme activity was not enhanced by supplementing FAD to the reaction. Together with the yellow colour of purified LhgO this indicates that bound FAD was co-purified saturating the reaction. Further, this activity of native LhgO in *E. coli* was found exclusively in the membrane fraction but not in the soluble protein fraction (Fig. 3a), and a Δ*lhgO* strain completely lacked the activity. Additionally, the redox activity of membranes upon L2HG addition was restored in a *lhgO+* complementation strain, but the activity was lower compared to WT *E. coli* membranes; this may be explained by inefficient incorporation of recombinant LhgO into the membrane, possibly caused by the His-Tag. Furthermore, we were able to show that the natural quinones of *E. coli* can serve as electron acceptors for LhgO. Membrane fractions of *E. coli* WT were incubated with L2HG or SA (testing succinate dehydrogenase as positive control) as substrates for the respective dehydrogenases. The ratio between ubiquinol to ubiquinone increased in the presence of both SA and L2HG compared to the control reaction without added substrate, indicating that electron transfer from these substrates to ubiquinone occurred in the membrane fractions (Fig. 3b). The ubiquinol/ubiquinone pool was unaffected by L2HG in a membrane fraction prepared from an *E. coli* strain lacking LhgO (Δ*lhgO*). We exclusively detected menaquinone and demethylmenaquinone in their oxidised but not reduced forms. However, many ETC-coupled dehydrogenases electron chain in *E. coli* show a wider substrate range for various quinone species and also under aerobic conditions, naphtoquinones can be used as electron acceptors[18,19]. Notably, since menaquinole and demethylmenaquinole reoxidise quickly[20] and might not be detectable in our assay, we cannot exclude that LhgO may accept

also these quinones as electron acceptors. However, when 2,6-chloroindophenol (DCPIP) was used as artificial electron acceptor both native LhgO in the membrane fraction and recombinant LhgO revealed higher activities in the presence of ubiquinone1 compared to menaquinone4 (Fig. 3c). Addition of the respiratory chain inhibitor 2-heptyl-4-hydroxyquinoline n-oxide (HQNO) in concentrations higher than 80 μM to membrane fractions abolished $O_2$ consumption and DCPIP activity, again confirming coupling of LhgO to the respiratory chain (Fig. 3d). Taking into account that for *E. coli* ubiquinone is the main electron acceptor under aerobic conditions[19], these results support the conclusion that LhgO acts as a quinone-dependent oxidoreductase located in the cytoplasmic membrane.

**The degradation of lysine to glutarate**. We next considered the source of GA, a compound that has not yet been shown to play an important and widespread role in metabolism. In *E. coli*, *csiD* and *lhgO* are co-encoded with genes for γ-aminobutyrate (GABA) shunt enzymes (*gabT* and *gabD*, Fig. 1). GabT is described as GABA transaminase that yields succinic semialdehyde. GabD is a dehydrogenase that converts succinic semialdehyde to SA[21]. We hypothesized whether GabT may convert also the C5-homolog 5-aminovalerate (AVA) to glutarate semialdehyde and whether GabD converts the glutarate semialdehyde to GA (Fig. 1), as it has been described for *Pseudomonas*[22] and *C. glutamicum*[23,24] homologues. We purified GabT and GabD from *E. coli* and demonstrated that they use AVA with comparable efficiency to GABA ($K_M^{GABA} = 197 \pm 27\ \mu M$, $K_M^{AVA} = 439 \pm 29\ \mu M$) for a coupled GabT/D reaction, (Fig. 4a). The reaction was additionally analysed via high-resolution mass spectrometry to confirm GA as the product derived from AVA (Supplementary Figure 8).

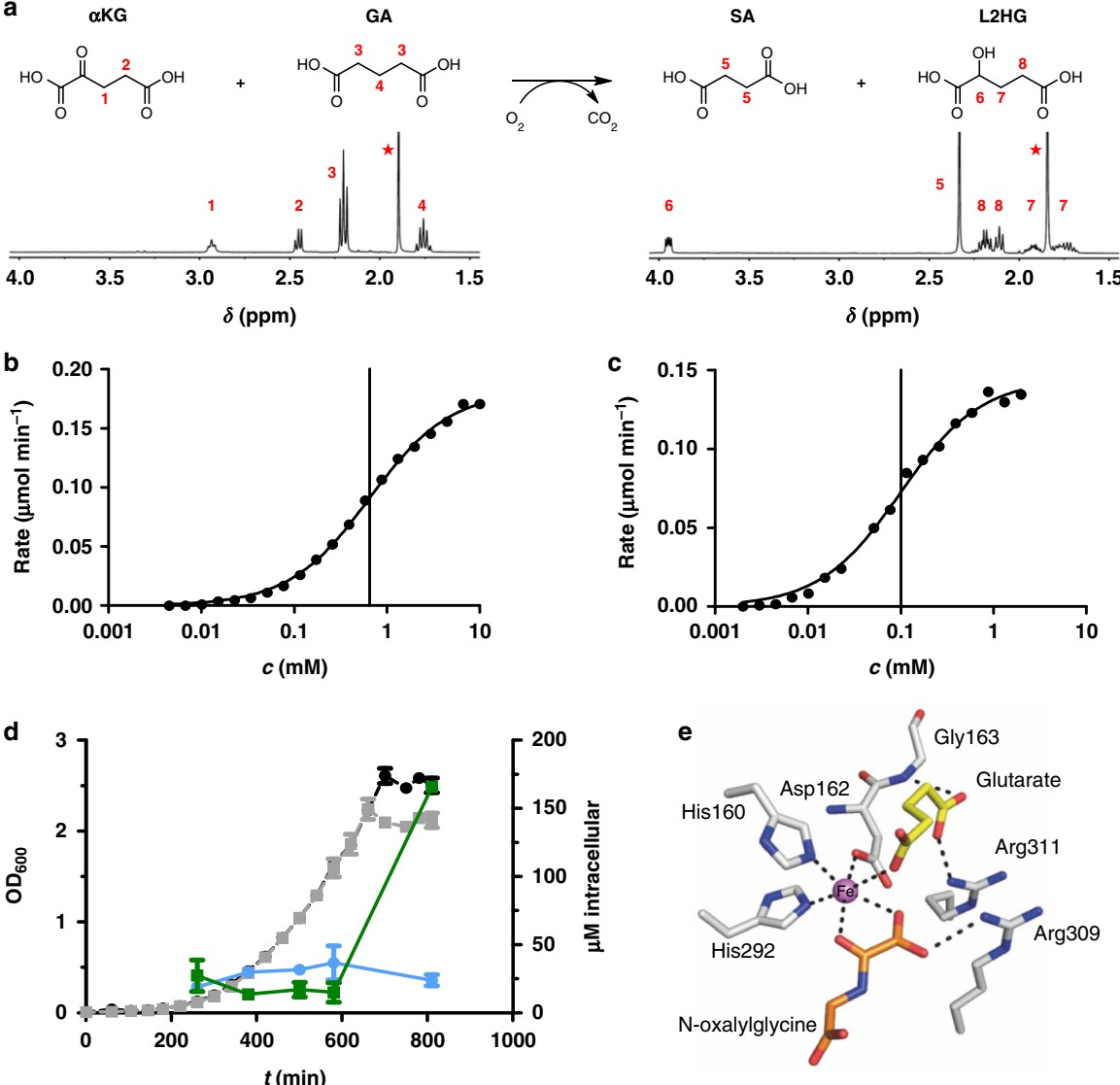

**Fig. 2** CsiD is an αKG-dependent dioxygenase converting GA to L2HG. **a** Reaction as catalysed by CsiD and its characterisation by NMR. CsiD converts GA and αKG to L2HG and SA, respectively (and carbon dioxide). [1]H-NMR spectra of educts (left) and products (right) after overnight incubation with CsiD in ammonium acetate buffer are shown, asterisks: signals from acetate buffer. **b, c** Reaction rate, measured as consumption of $O_2$ in a Clark-type oxygen electrode, is plotted against the concentration of GA (**b**) and αKG (**c**). Data were fitted with Michaelis-Menten equation (solid line). $K_M$ is indicated by the vertical line. **d** Intracellular GA concentrations in different growth phases of *E. coli* M9 cultures. Circles (WT *E. coli* strain) and squares (Δ*csiD E. coli* strain) represent intracellular GA concentrations (blue: WT, green: Δ*csiD*, right *y*-axis). Bacterial cell growth is indicated by optical density at 600 nm ($OD_{600}$) in black and grey (WT and Δ*csiD*, left *y*-axis). Shown are means of biological duplicates with error bars representing standard deviation including technical duplicates. **e** Crystal structure of CsiD. The active site of a protomer with substrate GA (yellow) and the αKG co-substrate analog N-oxalylglycine (orange) is shown. The model is a reconstruction obtained by superimposing individual crystal structures of CsiD bound to each of the ligands (the CsiD enzyme is equivalent in both structures; RMSD = 0.16 Å). The Fe ion is shown in magenta. Dashed lines represent interactions between ligands and residues in the binding pocket

For a potential source of AVA, we considered a pathway starting from lysine (Fig. 1), through decarboxylation of lysine to Cad followed by transaminase and dehydrogenase reactions. These reactions have been described in *Pseudomonas* species[3]. In *E. coli*, two lysine decarboxylases are known (CadA and LdcC)[1,2]. Reactions starting from the C4 diamine putrescine are catalysed by PatA and PatD, resulting in GABA[25–27]. It has already been shown that the *E. coli* transaminase PatA can process Cad with the same activity as putrescine[26]. Since there is no data available concerning PatD producing AVA in *E. coli*, we compared coupled PatA/D activities for putrescine and Cad as described for GabT/D before. The coupled PatA/D reaction with Cad showed a $K_M^{Cad}$ of $0.37 \pm 0.05$ mM and a $V_{max}$ of 11.8 μM min$^{-1}$, whereas PatA/D reaction with putrescine revealed a higher $K_M^{Put}$ of $1.43 \pm 0.07$ mM and a higher $V_{max}$ of 29.1 μM min$^{-1}$ (Fig. 4b). Furthermore, we confirmed the production of 5-aminopentanal (APA) and piperideine formed by PatA, as well as AVA by PatA/D via mass spectrometry analysis (Supplementary Figure 9). Our findings are further substantiated by the observation that overexpression of *ldcC*, *patA*, and *patD* from *E. coli* can be used for the production of AVA from lysine in *C. glutamicum*[28]. An alternative sequence of reactions that could produce AVA from lysine degradation is the putrescine utilization pathway (puuABCD) of *E. coli*, which is proposed to degrade extracellular putrescine[29].

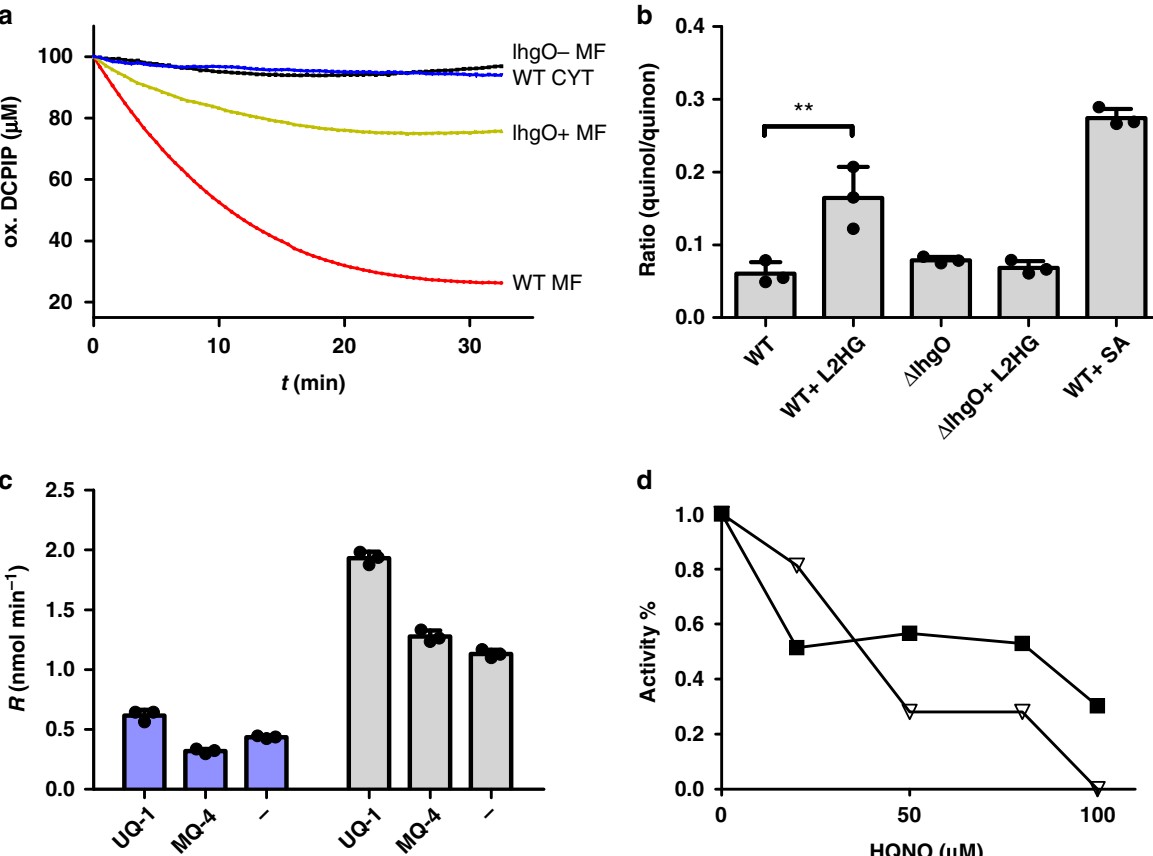

**Fig. 3** LhgO as a membrane-associated, L2HG:quinone oxidoreductase in *E. coli*. **a** Activity of *E. coli* WT membrane fraction (red; WT MF) transfering electrons from L2HG to DCPIP compared to a WT cytosolic fraction (blue; WT CYT), a membrane fraction isolated from a *ΔlhgO* knockout strain (black; *lhgO*– MF) and a complementation strain of *ΔlhgO* with recombinant LhgO (green, *lhgO*+ MF). Depletion of the oxidized form of DCPIP over time was measured photometrically. **b** Ubiquinol to ubiquinone ratio (red./ox.) was measured in *E. coli* membrane fractions in the presence or absence of L2HG or SA. Quinones were extracted from membrane fractions from either *E. coli* WT or *ΔlhgO* after the incubation with either L2HG or SA and measured by HPLC. Error bars represent standard deviations of triplicate experiments. Significance was assessed by a unpaired one-tailed Student's *t*-test (**$P$ value = 0.0083). **c** Reaction rate ($R$) of purified LhgO (blue columns) and *E. coli* WT membrane fraction (grey columns) measured by DCPIP reduction per time in dependence of ubiquinone 1 ($UQ_1$) or menaquinone-4 ($MQ_4$) after addition of L2HG, (-) no quinone added. Error bars represent standard deviations of triplicate experiments. **d** Inhibition of *E. coli* WT membrane activity by HQNO titration. HQNO was added before L2HG and rates were normalised to the activity without added HQNO. DCPIP reduction (spectrophotometrically) is indicated with black squares and oxygen consumption of membranes (Clark electrode) by white triangles

**Validation of the complete degradation pathway**. In order to validate the pathway in Fig. 1 we traced fully [13]C-labelled and [15]N-labelled lysine in growth experiments. *E. coli* was grown in minimal medium with glucose as carbon (and energy) source and supplemented with isotope-labelled lysine, since lysine cannot be utilized as the sole carbon source by *E. coli* (for growth experiments utilizing the intermediates of the lysine degradation pathway as C-sources and N-sources see Supplementary Figure 10). We found high intracellular concentrations of Cad, APA, and GA predominantly in the fully labelled form (Fig. 5a). Although GA was detected only at 180 μM in WT *E. coli*, GA levels accumulated to almost 5 mM in a *ΔcsiD* strain. Importantly, Cad, APA, and GA were found predominantly in their completely labelled form. Fully [13]C-labeled AVA was detectable at 640 μM in the *ΔgabT* strain. αKG and SA were detected predominantly in non-labeled form in accordance with multiple sources of these central metabolites. In the *ΔcsiD* strain we found highly elevated levels of [13]C-labeled GA but no [13]C-labeled metabolites downstream of the CsiD reaction (SA, αKG). In accordance with SA as end product of the lysine degradation pathway, [M+4] and [M+2]-labelled SA was observed in small amounts. The labelling pattern of the detected intermediates support an unbranched

pathway as depicted in Fig. 1. A detailed list of the intracellular concentrations and labelling patterns of all intermediates of the pathway can be found in extended data Table 3. Of note, we could also detect [13]C-labeled GA in a *ΔgabT* strain indicating that other transaminases besides GabT are able to process AVA. In *E. coli*, there are isoenzymes for both GabT and GabD known, namely the GABA transaminase PuuE and the succinate semialdehyde dehydrogenase Sad. We demonstrate for a coupled PuuE/Sad reaction with purified enzymes that similar to the findings with GabT/D (Fig. 4a) AVA can be used as a substrate (Supplementary Figure 11). Taken together, the isotope labelling experiments prove that lysine degradation as depicted in Fig. 1 is the source of GA in *E. coli*. It can be summed up as:

$$\text{L-lysine} + 2\,\alpha\text{KG} + 2\,\text{NAD(P)}^+ + O_2 + UQ \longrightarrow$$
$$SA + 2\,CO_2 + 2\,\text{glutamate} + 2\,\text{NAD(P)H} + UQH_2$$

**Characterisation of the allosteric repressor CsiR**. In *E. coli*, the *csiD* operon is followed by a transcription factor termed CsiR (Fig. 1) that has been shown previously to negatively regulate *csiD* operon expression[4]. Upon deletion of *csiR*, the expression of the

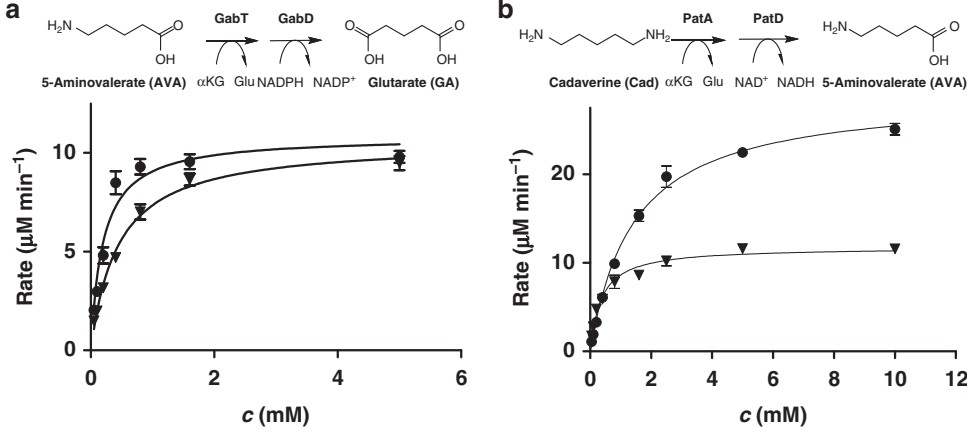

**Fig. 4** Promiscuous transaminase/dehydrogenase pairs involved in lysine degradation. **a** Coupled GabT (GABA transaminase) and GabD (succinic semialdehyde dehydrogenase) reaction: Comparison of the conversion of the C4 substrate GABA (circles) to SA with the C5 substrate AVA (triangles) to GA. **b** Coupled PatA (putrescine transaminase) and PatD (γ-aminobutyraldehyd dehydrogenase) reaction: Comparison of the conversion of the C4 substrate putrescine (circles) to GABA with the C5 substrate Cad (triangles) to AVA. Reaction rate (μM/min) is shown as a function of substrate concentration. Rate is given as μM $NADP^+/NAD^+$ reduced per min, monitoring the dehydrogenase activity. Data represent the mean of triplicate experiments with error bars representing standard deviations

*csiD* operon genes is induced. CsiR is homologous to the gluconate repressor[30] and hence predicted to be ligand-dependent. We purified CsiR from *E. coli* and utilizing a filter binding assay we demonstrate that the repressive effect at the *csiD* promoter site is specifically relieved upon GA binding, with related compounds showing no or less (αKG) effects (Fig. 5b, Supplementary Figure 12). We further examined the DNA binding site upstream of *csiD* by hydroxyl radical footprinting (Supplementary Figure 13a, b)[31]. We found that CsiR binds to two primary and two secondary sites in the promoter region of the *csiD* operon with the consensus sequences $TTGTN_5TTTT$ and $ATGTN_5TTTT$ of the primary sites, each separated by six nucleotides (Fig. 5c). Surface plasmon resonance studies determined a dissociation constant $K_d$ = 10 nM for the CsiR/DNA interaction (Supplementary Figure 14a, b).

## Discussion

Lysine is originally thought to be degraded in a ketogenic manner via β-oxidation and the formation of acetyl-CoA. In accordance with that, GA is an intermediate of lysine degradation in Pseudomonads and is proposed to be converted to glutaryl-CoA which is further metabolized to acetyl-CoA[32]. Recently, a study of lysine degradation in *P. putida* KT2440 was published that identified and characterized the same pathway via CsiD and LhgO as in this work[33]. The authors also demonstrated that *E. coli* possesses the key enzymes CsiD and LhgO necessary to degrade lysine via GA and L2HG. Hence, in *E. coli*, lysine is metabolised in a glucogenic way by the conversion of GA to SA and $CO_2$ via αKG-dependent hydroxylation and subsequent oxidation to αKG. This pathway fills a gap in central carbon and energy metabolism. In *E. coli*, the pathway is activated in the stationary phase in a cAMP/CRP-dependent manner. Hence, it seems that lysine is recycled by *E. coli* in stationary phase for carbon and energy regeneration. The described regulation is in accordance with growth experiments with the *ΔcsiD* strain exhibiting a growth defect at the onset of the stationary phase compared to WT *E. coli* K-12 (Supplementary Figure 15), hinting at a distinct advantage of activating lysine degradation via the *csiD* pathway in the stationary phase. Apart from introducing reactions of central metabolites, our findings have implications for the biotechnological production of the polyamide building

blocks AVA and GA. Both compounds have been produced from engineered *E. coli* before[34,35]. Taking into account the presented pathways and activities, it is likely that yields could be improved significantly by utilizing knockout strains or adopting conditions such as oxygen limitation preventing degradation of the target compounds by GabT/D and CsiD/LhgO. Indeed, Zhang and co-workers demonstrate increased production of GA by knocking out both the glutarate hydroxylation as well as the glutaryl-CoA dehydrogenase pathway in *P. putida*[33].

Furthermore, the presented pathway has potential implications for diseases such as cancer[9] and genetic organic acidurias[36]. In humans, L2HG is produced by malate and lactate dehydrogenases especially at acidic conditions caused e.g., by hypoxia[37]. Increased L2HG levels are malignant due to inhibition of TET-type and Jmjc-type αKG-dependent oxygenases responsible for nucleobase and histone demethylation, resulting in epigenetic deregulation of gene expression and thereby progression in certain cancers[13,14]. Some tumours are also more prevalent in genetic acidurias defined by elevated levels of L2HG[38]. Genetic inactivation of L2HGDH (the human homologue of LhgO) is the cause of *L*-hydroxyglutaric aciduria[39]. Increased GA levels are found in glutaric aciduria type I[40] and glutaric aciduria type III[41]. The glutaric acidurias are caused by malfunctioning glutaryl-CoA dehydrogenase and succinyl-CoA/glutaryl-CoA transferase, respectively[41]. However, the source of free GA in human metabolism is not fully understood since interrupting the saccharopine pathway upstream of glutaryl-CoA formation in a mouse model for GA-I did not rescue the mice from developing the disease[42]. Moreover, certain cases of glutaric aciduria responded to antibiotics treatment, hinting at the possibility that gut bacteria could contribute to elevated levels of GA[43]. Interestingly, questions regarding additional sources of GA and hydroxyglutarates and their potential involvement in normal metabolic processes have been raised frequently[9]. In this regard, the presented work is potentially relevant for the discussed diseases and could inspire further research identifying sources and fates of GA and L2HG in humans.

## Methods

**Enzyme nomenclature**. We suggest that the discovered glutarate hydroxylase belongs to subgroup EC 1.14.11. with the name glutarate-2-hydroxylase (G-2-H)

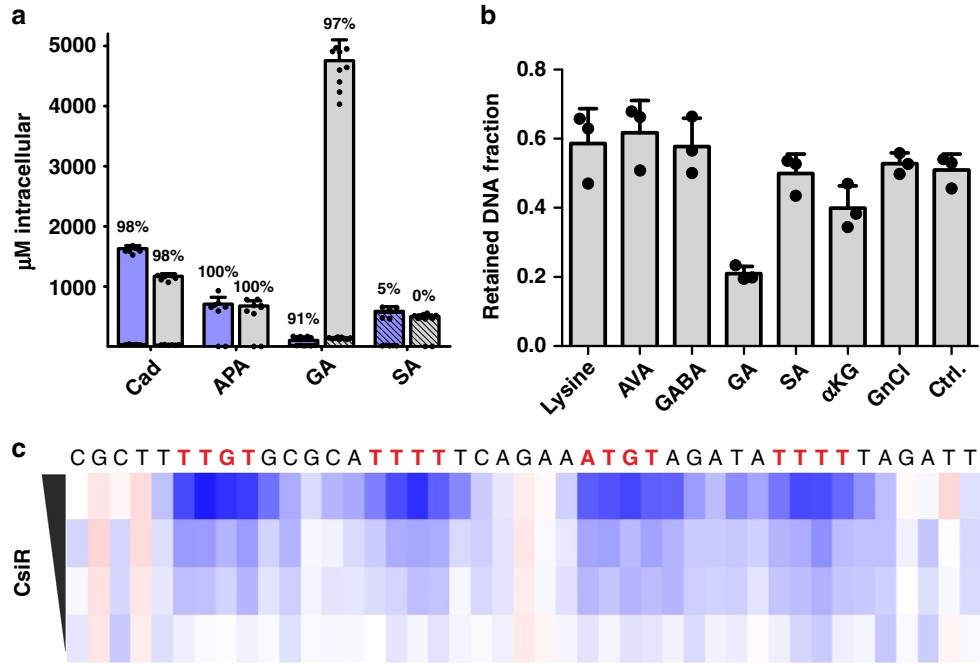

**Fig. 5** Isotope tracing and regulation of the lysine degradative pathway in *E. coli*. **a** Intracellular concentrations of cadaverine (Cad), 5-aminopentanal (APA), glutarate (GA) and succinate (SA) of *E. coli* K-12 wt (blue) and *ΔcsiD* (grey) in stationary phase cultures. Non-isotope-labelled compounds are indicated as shaded bars. The sum of $^{13}C$-labeled metabolites are shown as blank bars. Fractions of labelled compounds relative to the sum of labelled and non-labelled compounds are indicated above the respective bar (in percent). For isotope labelling patterns see Supplementary Table 3. Data represent the means of biological triplicates with error bars representing standard deviations. Each replicate was measured at least once. **b** characterisation of the allosteric repressor CsiR of the *csiD* operon: Ligand specificity of CsiR repression determined by a filter binding assay (Supplementary Figure 8) of CsiR and a dsDNA sequence derived from the *csiD* operon promoter region in presence of annotated compounds (1 mM each, GnCl: Guanidiniumchloride, Ctrl: no ligand). CsiR-bound radioactive DNA is retained on a nitrocellulose membrane. GA specifically reduces binding of CsiR to the DNA. and **c** determination of dsDNA consensus motif recognized by CsiR determined by hydroxyl footprinting (Supplementary Figure 13a, b). Increasing CsiR concentrations protect DNA from hydroxyl radical cleavage (decreased cleavage is visualised in blue)

and systematic name glutarate, 2-oxoglutarate:oxygen oxidoreductase ((2S)-hydroxylating). We propose to rename the *csiD* gene to glutarate hydroxylase (*glaH*). LhgO should be reclassified as L-2-hydroxyglutarate dehydrogenase belonging to EC 1.1.99.2 (systematic name: (S)-2-hydroxyglutarate:quinone-oxi-doreductase) instead of 1.1.3.15 ((S)-2-hydroxy-acid oxidase) and the gene should be renamed from *lhgO* to *lhgD* in analogy to the homologous dehydrogenases. The gene encoding the HTH-type transcriptional repressor could be renamed from *csiR* to *glaR*.

### Bacterial strains and general culture conditions.

*Escherichia coli* BW25113 and its single gene knockout derivatives *ΔcsiD*, *ΔgabT*, *ΔglcD*, and *ΔlhgO* from the Keio collection[44], that were used for analysis, were all obtained from the National BioResource Project (NIG, Japan): *E. coli*. Genes encoding CsiD, CsiR, GabD, GabT, LhgO, PuuE, Sad were amplified and restriction enzyme sites were introduced with primers listed in Supplementary Table 4. *Escherichia coli* BW25113 was used as template. PCR fragments were inserted into vector pET28a (Km$^r$) carrying a T7-promoter under control of *lacI* using standard protocols. Thus, over-expression vectors were created for CsiD, CsiR, GabD, GabT, PuuE, and Sad ORFs carrying a 6× N-terminal His-tag and for LhgO carrying a 6× C-terminal His-tag. To create a complemented *lhgO+* strain, *lhgO* was inserted into pET16b using restriction sites and primers listed in Supplementary Table 4. Recombinant cloning vectors were transformed into *Escherichia coli* BL21 (DE3) (Stratagene, USA, La Jolla) via electroporation. Plasmid constructs were validated by sequencing using primers SP04 and SP10. Bacterial strains were either grown in LB medium or M9 minimal medium containing 8.5 g/l Na$_2$HPO$_4$·2H$_2$O, 3 g/l KH$_2$PO$_4$, 0.5 g/l NaCl, 1 g/l NH$_4$Cl, 2 mM MgCl$_2$, 100 μM CaCl$_2$ supplemented with trace elements (0.1 mM EDTA, 0.03 mM FeCl$_3$, 6.2 μM ZnCl$_2$, 0.76 μM CuCl$_2$, 0.42 μM CoCl$_2$, 1.62 μM H$_3$BO$_3$; 0.08 μM MnCl$_2$) and vitamins (0.1 mg/l cyanocobalamin, 0.08 mg/l 4-aminobenzoic acid, 0.02 mg/l D-(+)-biotin, 0.2 mg/l niacin, 0.1 mg/l Ca-D-(+)-pantothenic acid, 0.3 mg/l pyridoxamine-chloride, 0.2 mg/l thiamindichloride) at 37 °C. As carbon source in minimal medium 0.2% (w/v) glucose was used. When necessary, medium was supplemented with 30 μg/ml kanamycin.

### Growth on different C-sources and N-sources.

*E.coli* strains were grown in 96 deep well plates in minimal medium as triplicates containing trace elements supplemented with 10 mM C-sources (lysine, cadaverine, 5-aminovalerate, glutarate) or N-Sources (lysine, cadaverine, 5-amniovalerate). Growth was monitored by measuring OD$_{600}$ in 96 well plates by Tecan plate reader. Growth (means of triplicates) was rated as high growth (0.47 > OD$_{600}$ > 0.43 a.u.); intermediate growth (0.35 > OD$_{600}$ > 0.25 a.u.), low growth (0.15 > OD$_{600}$ > 0.07 a.u.) or no growth (0.02 a.u > OD$_{600}$).

### Protein purification.

Strains were grown in LB with suitable antibiotic to an OD$_{600}$ of 0.5. Protein expression was induced with 1 mM IPTG and cells were grown for 4–5 h. Cells were harvested by centrifugation at 4000 rpm at 4 °C. Cells were lysed by sonication in lysis buffer (50 mM NaH$_2$PO$_4$, 300 mM NaCl, 10 mM imidazole) with Branson sonifier for 3 min duty cycle at 20% power with 0.5 on and 0.5 s off. Cell debris was removed by centrifugation at 10,000 rpm at 4 °C and the cell lysate incubated with Ni-NTA beads (Qiagen) for 1 h at 4 °C at a rotary shaker. Beads were washed twice with wash buffer (50 mM NaH$_2$PO$_4$, 300 mM NaCl, 20 mM Imidazole) and the protein finally eluted with elution buffer (50 mM NaH$_2$PO$_4$, 300 mM NaCl, 500 mM Imidazole). Purity was checked by SDS-gel. Protein was transferred to the respective buffer system required for the experiment by size exclusion chromatography (PDE-10 columns, GE healthcare).

### NMR of the CsiD reaction.

The reaction of α-ketoglutarate (10 mM) and glutarate (10 mM) catalysed by CsiD in 20 mM ammonium acetate buffer pH 7.25 was monitored by $^1$H NMR spectra without purification. $^1$H NMR spectra were recorded at 300 K on Bruker Avance III 400 MHz spectrometers (ayita 400 and isa 400) with a BBFO plus probe for N to F/H or F. Data for NMR spectra were recorded as follows: chemical shift (δ, ppm), multiplicity (s, singlet; d, doublet; t, triplet; q, quartet; m, multiplet), integration, coupling constant (Hz). Acquired data was processed and analysed using MestReNova software.

### CsiD reactions analysed by Clark-type O$_2$ electrode.

Reactions were conducted in 100 mM MOPS, 70 mM NaCl and 20 mM KCl pH 7.2 in the presence of 400 μM Ascorbat and 4 μM Fe(NH$_4$)$_2$SO$_4$. $K_M$ for glutarate and α-ketoglutarate was determined in the presence of 1 mM α-ketoglutarate and 5 mM glutarate, respectively. Five millimolar substrate was used for the C2–C7 analogues of glutarate. 180 nM CsiD were used for the reactions. Reactions were conducted at 30 °C in the sealed reaction chamber with magnetic stirrer. $v_0$ was determined and plotted

against the concentration. The plot was fitted with Michaelis-Menten equation with in GraphPad Prism5 (HYPNOS).

**Stereospecificity of the CsiD reaction**. Evaluation of the 2-hydroxyglutarate enantiomer was performed as described before[12]. The CsiD reaction was performed as before (oxygen electrode) for 1 h at 30 °C. Reaction was incubated for 10 min at 70 °C to heat-inactivate CsiD. Denatured protein was removed by centrifugation (14,000 rpm). Supernatant of the CsiD reaction and L-2-hydroxyglutarate and D-hydroxyglutarate as controls were derivatised with 50 g/L diacetyl-L-tartaric anhydride in 4:1 (v/v) dichloromethane/acetic acid for 30 min at 75 °C. Supernatant was evaporated to dryness and residue was dissolved in water. Derivatised products were analysed by LC-MS and identified by comparison to standards.

**Crystallisation of CsiD-ligand complexes**. Purified CsiD protein was concentrated to 13 mg/mL and its crystallization in liganded form pursued by co-crystallization, crystal soaking or a combination of both approaches, as follows:

Apo CsiD: crystals grew from reservoir solutions containing 80 mM sodium chloride, 12 mM potassium chloride, 20 mM magnesium chloride hexahydrate, 40 mM sodium cacodylate trihydrate pH 6.0, 30% [v/v] 2-methyl-2,4-pentanediol, 12 mM spermine tetrahydrochloride.

CsiD-GA: crystals from apo CsiD (obtained as described in i) were soaked in 10 mM glutarate prior to flash cooling.

CsiD-SA: CsiD was co-crystallised with succinate by using a mother liquor containing 1.0 M succinic acid pH 7.0, 0.1 M Bis-Tris propane pH 7.0.

CsiD-NOG: CsiD protein was incubated with NOG at a CsiD:NOG molar ratio of 1:5 for 1 h on ice and the mixture used in crystallisation trials. Crystals grew from 1.0 M magnesium sulfate, 0.1 M Tris pH 8.5. Prior to flash freezing, crystals were soaked in 20 mM NOG to improve ligand occupancy.

In all cases, crystallization was performed in 96-well Intelliplates (Art Robbins) and used the sitting drop, vapour diffusion format implemented with a Gryphon (Art Robbins) nanolitre dispensing robot. Drops consisted of 200 nL protein or protein/ligand mixture and 200 nL reservoir solution, with reservoirs containing 70 μL mother liquor. All crystallization trials were incubated at 18 °C. For X-ray data collection, crystals were harvested into LithoLoops (Hampton Research) and cryo-protected in mother liquor supplemented with 20% [v/v] glycerol.

**X-ray data collection and structure elucidation**. X-ray diffraction data were collected at beamline PXI (X06SA) of the Swiss Light Source synchrotron (Villigen, Switzerland) equipped with an Eiger 16 M detector (Dectris, Switzerland). Data were collected at a wavelength of 1.00 Å with 0.1°–0.2° oscillation per frame. Data processing used the XDS/XSCALE package[45]. Phasing was performed by molecular replacement in PHASER[46]. First, a CsiD protomer obtained from PDB entry 2R6S[15] [10.2210/pdb2R6S/pdb] was used as search model for the elucidation of the apo CsiD structure in this work. The resulting apo model was then used for the phasing of liganded crystal forms of CsiD. Model refinement used phenix.refine[47] and manual model building was performed in COOT[48]. Ligand restraints were generated using ELBOW[49]. Non-crystallographic symmetry restraints and TLS refinement were applied as part of model refinement in phenix.refine[47]. All structures had good Ramachandran values (Favoured: >98%; Disallowed: 0%). X-ray data statistics and model parameters are given in Supplementary Table 1. (Diffraction data and model coordinates have been deposited with the Protein Data Bank. Accession codes are given in Supplementary Table 1).

**Kinetics of GabT/D and Sad/PuuE and PatA/D**. Assays were carried out in a buffer (pH = 8.0) containing 5 mM α-ketoglutarate, 500 μM NADP+ (GabT/D & Sad/PuuE) or NAD+ (PatA/D), 100 μM DTT, 100 mM sodiumpyrophosphat and 0.01 mg/mL of both purified GabT/D, Sad/PuuE or PatA/D. Reactions were conducted at room temperature and started by the addition of different concentrations of AVA or GABA (GabT/D & Sad/PuuE) respectively cadaverine or putrescine (PatA/D). Increase of NADPH or NADH was monitored over time measuring the absorbance at 340 nm, thus determining the kinetic of the coupled enzyme reaction. The reduction of NADP+ to NADPH corresponds stoichiometrically to the conversion of GABA (or AVA) to succinate semialdehyde (or glutarate semialdehyde) and then to succinate (or glutarate) (GabT/D). In case of PatA/D, NAD+ reduction corresponds to the conversion of putrescine (or cadaverine) to amino-butanal (or aminopentanal) followed by oxidation to GABA (or AVA) catalysed by PatD. Decoupling the assay by first starting the transaminase reaction followed by addition of the dehydrogenase did not alter overall velocities concluding that the dehydrogenase reaction must be the rate limiting step in the reaction. Starting velocities ($v_0$) of the reaction were plotted against substrate concentrations and non-linear regressions were calculated with GraphPad Prism 5 (HYPNOS).

**Preparation of E. coli membrane fractions**. Wildtype and knockout E. coli strains were inoculated in 200 mL LB medium and incubated over night at 37 °C at 200 rpm on a rotary shaker. Overnight culture was centrifuged and the pellet was washed twice with ice-cold membrane fraction buffer containing 50 mM HEPES, 10 mM potassiumacetate, 10 M CaCl2, 5 mM MgCl2 titrated with NaOH to pH = 7.5 and resuspended in 10 mL buffer. All following steps were carried out on ice or

at 4 °C. Cell suspension was passed four times through a french press cell at 10,000 psi. To remove unlysed cells and insoluble cell debris the lysate was centrifuged at 20,000×g for 20 min. Supernatant was centrifuged for 30 min at 100,000×g. After ultracentrifugation supernatant was stored as cytosolic fraction and the brownish membrane pellet was washed with membrane fraction buffer (5 mL) and centrifuged again. The membrane pellet was resuspended in 500 μL membrane fraction buffer. Protein concentration of cytosolic and membrane fraction was determined via BSA-Bradford assay.

**Activity of purified LhgO and membrane fractions**. Prior to analysis a PD midiTrap G-25 column (GE Healthcare) was used to exchange buffer of purified LhgO to retain the protein in reaction buffer containing 25 mM HEPES, 100 mM NaCl, 5 mM EDTA (pH = 7.5). LhgO activity was assayed spectrophotochemically in reaction buffer. Direct reduction of 100 μM ubiquinone-1 (UQ1) (coenzyme Q1, Sigma Aldrich Co.) by purified LhgO was measured by the decrease of absorbance at 278 nm. A final enzyme concentration of 650 nM was used and the reaction was monitored dependent on L-2-hydroxyglutarate concentration. Differential absorption coefficient of UQ1 at this wavelength in reaction buffer was determined as $\Delta\varepsilon = 8.36$ mM. Enzyme activities were determined as UQ1 reduced per minute. Specific activities were referred to 0.03 mg/mL LhgO used in the assay. To exclude possible effects of H2O2 produced by LhgO on redox dyes 100 U/mL catalase (Catalase from bovine liver, Sigma Aldrich Co.) was added. Initial velocities ($v_0$) of the reaction were plotted against L2HG concentrations and non-linear regressions were calculated with GraphPad Prism 5 (HYPNOS).

Redox activity of 0.1 mg/mL total protein of membrane-fraction and cytosolic-fraction was determined in membrane fraction buffer (50 mM HEPES, 10 mM potassiumacetate, 10 mM CaCl2, 5 mM MgCl2, pH = 7.5). Reduction of 100 μM DCPIP by purified LhgO (217 nM) or membrane-bound enzyme was monitored in presence or absence of 50 μM ubiquinone and menaquinone-4 (MQ4) (menaquinone K2, Sigma Aldrich Co.) by absorbance measurement at 600 nm. Oxidized DCPIP has an absorbance maximum at 600 nm. Activities of membrane-bound and purified LhgO were determined as DCPIP reduced per minute (nmol/min). DCPIP reduction of E. coli WT membranes compared to wildtype cytosolic fraction and membranes isolated from a ΔlhgO strain were conducted in the presence of 50 μM UQ1. DCPIP reduction assays were carried out at room temperature and started by the addition of 5 mM L-2-hydroxyglutarate.

Oxygen consumption by membranes and purified LhgO were measured in a Clark-type electrode at room temperature. Activity of membrane-bound LhgO upon titration of the respiratory chain inhibitor HQNO was determined by oxygen consumption and DCPIP reduction.

**Bacterial ubiquinone reduction**. Membrane fractions of E.coli WT and ΔlhgO were prepared as described in the section "Preparation of E. coli membrane fractions". In brief, cells were lysed in a french press and membranes were isolated by ultracentrifugation. Total protein concentration was determined. One hundred and fifty microgram of the membrane fraction in 200 μL membrane fraction buffer (50 mM HEPES, 10 mM potassiumacetate, 10 mM CaCl2, 5 mM MgCl2, pH = 7.5) were incubated for 1 h at 25 °C under nitrogen atmosphere in the presence or absence of 5 mM L2HG or SA. Quinones were extracted with 1 mL of a 1:3:1 mixture of methanol:hexane:acetone. Samples were dried under a constant flow of nitrogen and analysed by HPLC coupled to a PDA detector (Shimadzu) on a Eurospher RP C18 column (125 × 3 mm; Knauer). Mobile phases contained 100% methanol (A) and 10% heptane in methanol (B). Isocratic elution with 100% A was performed for 10 min followed by a linear gradient from 0 to 50% B in 2 min and 50 to 100% B in 12 min. Spectra were recorded from 200 nm to 400 nm. Area under the curve were determined for ubiquinol (RT: 9 min; 290 ± 4 nm) and ubiquinone (RT 16 min; 275 ± 4 nm). Statistical significance of the data was assessed by an unpaired one-tailed Student's t-test in GraphPad Prism5 (HYPNOS). The nature of the quinone species and reduction state was confirmed by the corresponding retention times and absorption spectra[50–52]. Furthermore, ubiquinone from the membrane fractions could be assigned to the mass of ubiquinone-8 by mass spectrometry.

**Growth experiments with isotopically labelled lysine**. For growth and metabolite analysis of Escherichia coli BW25113 WT and ΔcsiD cells were inoculated in 100 mL M9 minimal medium supplemented with glucose in the presence of 10 mM lysine in 500 mL baffled flasks. Cultures were grown in duplicates at 37 °C on a rotary shaker at 200 rpm. Cell growth over time was determined spectro-photochemically by measuring optical density at 600 nm in a 1 mL cuvette at indicated intervals. For monitoring intracellular metabolite concentrations during E. coli culture growth samples with a volume equivalent to OD600 = 0.1/mL were taken at indicated time points.

For [13]C labeling experiments WT and ΔcsiD cells were inoculated in triplicates supplemented with 10 mM L-lysine-[13]C6, [15]N2 hydrochloride (Sigma Aldrich Co.) in 5 mL M9 minimal medium containing 0.2% (w/v) glucose in 50 mL falcons and incubated at 37 °C for 24 h at 200 rpm. After 24 h a volume equivalent to OD600 = 0.2/mL was harvested and centrifuged at −9 °C. To rapidly quench metabolic activities the harvested cells were methanol-extracted by resuspending the cell pellet in 2 mL 80% (v/v) methanol cooled down close to freezing with liquid

nitrogen. The suspension was frozen in liquid nitrogen for 1 min, thawed on ice, centrifuged for 10 min at 10,000×*g* and the supernatant was stored as metabolite extract. This step was repeated three times. The supernatant fractions were pooled and lyophilized to dryness. Lyophilized metabolite extract was dissolved in 500 μL deionized water and analyzed by LC-MS.

**Analysis of lysine degradation metabolites by LC-MS**. Identification of metabolites were performed with UltiMate 3000 HPLC system and LTQ Orbitrap Velos (Thermo Scientific). Nucleodur C18 ISIS column (250 mm length x 2 mm i.d., 2.7 μm particle size, Macherey-Nagel, Germany, Düren) was used. The injection volume was 10 μL and the flow rate was 0.25 mL min$^{-1}$. The mobile phases contained 10 mM ammonium formate (pH 3.2) (A) and 0.1% formic acid in acetonitrile (B). The linear gradient comprised 0 to 30% B for 10 min and 90% B for 2 min. The MS scan ranged from an *m/z* 100–400 with a resolution of 100,000 at *m/z* 400 was achieved in positive and negative ionization modes. Accurate mass (±3 ppm) and retention time (±0.2 min) values were used for molecular assignment. For the analysis of hydroxyglutarate derivatized with DATAN Prominence HPLC system with LCMS-2020 single quadrupole MS (Shimadzu) was applied. HPLC conditions was as previously described. MS Detection was performed in single ion monitoring (SIM) negative ionization mode at *m/z* 363. The same instrumentation was used for the quantification of metabolites in cell extracts. Prior to analysis 16 μL of aqueous metabolite extract was mixed with 8 μL mobile phase B (90% acetonitrile, 0.2% formic acid, 10 mM ammonium formate), 5 μL of this solution was injected. Metabolites were separated using Nucleodur HILIC column (250 mm length x 2 mm i.d., 3 μm particle size, Macherey-Nagel), which was equilibrated with buffer B and eluted with a linear gradient of 45% buffer A (10 mM ammonium formate, pH 3.0) over a 10-min period followed by isocratic step of 45% buffer B for 8 min. Column was operated at (35.0 ± 0.1) °C with a flow rate of 0.15 ml/min. SIM detection of corresponding protonated ions in positive ionization mode was used for lysine, cadaverine and 5-aminovalerate. Glutarate, succinate and α-ketoglutarate ions were detected in negative SIM mode. Ions used for identification and quantification of compounds by LC-MS are summarised in Supplementary Table 2.

To identify and quantify compounds in cell extracts standard solutions of pure substances were measured under the same conditions. Calibration curves for cadaverine, glutarate and succinate were obtained for the measuring range and used for quantification of these substances. Intracellular metabolite concentrations were calculated according to following equation:

$$C_{avg} = C_{ex} \times V_{ex} \times \frac{DW_{cell}}{DW_{tot} \times V_{cell}}, \qquad (1)$$

with $C_{ex}$ being the metabolite concentration of the extraction solution determined via external calibration. $V_{ex}$ is the volume of the extraction solution (0.5 mL). $DW_{tot}$ is the experimentally determined total dry weight of the metabolite sample. $DW_{cell}$ is the dry weight per cell ($3 \times 10^{-13}$ g) and $V_{cell}$ is the volume per cell ($6.7 \times 10^{-16}$ L)[53].

**Filter binding of CsiR/dsDNA interaction**. Primer MS161 was 5′-labelled with γ-P32-ATP (Hartmann Analytics) by T4 PNK (NEB). PCR was conducted with labelled MS161 and MS162 on genomic DNA from *E. coli* K-12 MG1655. PCR product was purified by agarose gel extraction. DNA (500 cps) was incubated for 15 min in the presence of purified CsiR and 1 mM compound (as assigned) in binding buffer (50 mM Tris-HCl, 100 mM KCl, 50 mM NaCl, 5 mM MgCl$_2$ pH = 7.6). Reaction was dot-blotted on nitrocellulose membrane (0.2 μM, Roth) followed by a nylon membrane (0.45 μM, Roth). Binding was determined by the ratio of bound DNA (Intensity on NC membrane) to entire DNA (Sum of Intensity on NC membrane and nylon membrane). Radiograph was recorded with Thypoon FLA 7000 (GE Healthcare).

**Hydroxyl radical footprint**. Primer MS170 was 5′-labelled with γ-P32-ATP (Hartmann Analytics) by T4 PNK (NEB). PCR was conducted with labelled MS170 and MS171 on genomic DNA from *E. coli* K-12 MG1655. Binding of CsiR was allowed for 15 min at 25 °C in 50 mM MOPS, 100 mM NaCl, 20 mM KCl, 2.5 mM MgCl$_2$ 0.1 mM DTT pH = 7.2. Footprint reaction was performed as described before[31]. Additionally, a G specific Maxam-Gilbert sequencing reaction was performed as a control to assign the nucleotides. DNA fragments were separated on a 10% denaturing PAGE. Radiograph was recorded with Thypoon FLA 7000 (GE Healthcare). The radiograph was analysed with SAFA Quant[54].

**SPR of CsiR/DNA interaction**. Part of the *csiD* 5′-UTR containing the promoter region was amplified by PCR with primers MS178 and biotin-tagged MS177. The DNA was immobilized in TES buffer (10 mM Tris, 500 mM NaCl, 1 mM EDTA pH = 7.6) to reach 300 RU. Purified CsiR was diluted in running buffer (50 mM MOPS, 100 mM NaCl, 20 mM KCl, 2.5 mM MgCl$_2$, 0.1 mM DTT, 5 μg/mL sheared Salmon sperm DNA, 0.1 mg/mL BSA and 0.005% Tween20 pH = 7.2). Binding for indicated CsiR concentration was examined at a flow rate of 30 μL/min. The surface was regenerated with a short pulse of 0.05% SDS. SPR was performed on a Biacore T200. The resulting sensograms were analyzed using the Biacore

Evaluation Software. Binding of 0.9 μM CsiR was fitted kinetically assuming a 1:1 interaction.

## Data availability

Data supporting the findings of this manuscript are available from the corresponding author upon reasonable request. A reporting summary for this Article is available as a Supplementary Information file. The presented crystal structures are available as PDBs 6GPE, 6HL8, 6GPN, and 6HL9.

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

## Acknowledgements

We thank Astrid Joachimi, Anna Heiler, Yuanhao Li, Dennis Kläge, Vera Hedwig, Malin Bein, Christoph Globisch, and the members of the Proteomics and NMR facilities of University of Konstanz for technical assistance and helpful discussions. This work was supported by an ERC Consolidator grant to J.S.H. We thank the Swiss Light Source synchrotron (Villigen, CH) for access.

## Author contributions

S.K., M.S., and J.S.H. conceived and designed this study. S.K., M.S., D.G., C.W., and D.S. analysed the CsiD reactions. R.M.W. and O.M. crystallized and solved the CsiD structures. S.K. and N.M. characterized the LhgO reaction. S.K., M.S., and D.G. analysed the GabT/D, Sad/PuuE, and PatA/D reactions and carried out the isotope tracing experiment. M.S. analysed the ligand-dependency of CsiR. S.K., M.S., and J.S.H. wrote, and all authors commented on the manuscript.

## Additional information

**Competing interests:** The authors declare no competing interests.

