## [Peer Review File · Nature Communications]

Reviewers' Comments:

Reviewer #1:

Remarks to the Author:

Review comments for Nature Communications manuscript NCOMMS-18-19566-T, Wide-spread bacterial lysine degradation proceeding via glutarate and L-2-hydroxyglutarate

Major comments:

The authors report a bacterial lysine degradation pathway that proceeds via glutarate and L-2-hydroxyglutarate with CsiD and LhgO as the key enzymes in *Escherichia coli*. Unfortunately, the pathway is essentially the same as that in *Pseudomonas*, which also involves CsiD and LhgO (Nat Commun, 2018, 9:2114). It is likely that the authors did not read the paper since the paper was not cited in the manuscript. In comparison, this manuscript (Knorr et al.) reports the functions of CsiD, LhgO, GabT, GabD and CsiR in *E. coli*. In previous works, the functions of CsiD, GabT and GabD have been identified in *Pseudomonas putida* (Nat Commun, 2018, 9:2114; Appl Environ Microbiol, 2001, 67:5219-24) and the functions of LhgO and CsiR have been identified in *E. coli* (J Bacteriol, 2008, 190:3793-3798; Mol Microbiol, 2004, 51:799-811). The only new information in this manuscript is related to CsiR. Thus, this manuscript significantly overlaps with the previous reports. Thus, the novelty of Knorr et al.'s manuscript is not enough to be published in a distinct journal such as Nature Communications.

Specific Comments:

1. This reviewer is fully convinced of the need of *in vivo* studies providing the proposed roles of the enzymes studied in this work. Specially, the growth of every mutant in M9 medium containing lysine, 5-aminovalerate, glutarate, and L-2-hydroxyglutarate should be assayed.
2. In a previous work (J Bacteriol, 2008, 190:3793-3798), LhgO was identified as an oxidase. In this work, the same protein was identified as a ubiquinone-linked dehydrogenase. The authors should do more experiments to support their claims.
3. Since the regulation mechanism of CsiR is the only new information in the manuscript, the authors should pay more attention on this part. Some important experiments such as EMSA, lacZ reporter fusions should be conducted. The results of footprinting should also be further confirmed by EMSA.
4. In abstract, what is the meaning of "dead-end metabolite"? Glutarate can be metabolized to acetyl-CoA through a series of reactions. It is not suitable to define glutarate as a "dead-end metabolite".
5. In abstract section the authors introduced possible relation between gut microbiome and the chemicals studied in this manuscript. However, there was no detailed discussion about this issue in the main text. No data was involved to support the view in the abstract "Our work introduces the gut microbiome as a potential source of these metabolites associated with human diseases such as cancer and organic acidurias".
6. The abstract, Figure 1 and the legend should be revised to highlight the regulation mechanism of CsiR instead of the identification of the pathway.
7. Correspondingly, extended Data Fig. 8, Fig. 9 and Fig. 10 might be better to locate in the main text to characterize CsiR.
8. Site-specific mutagenesis is required to determine the key sites involved in the regulation process of CsiR.
9. To demonstrate "LhgO is the only L-2-hydroxyglutarate-converting enzyme in *E. coli*", on the basis of the results of enzyme assay of Δ LhgO, the author might add some growth experiments with lysine or glutarate as the sole carbon source and thus determine the intracellular change of L-2-hydroxyglutarate.
10. Some key experiments based on the knockout strains are better to conduct again with gene-complementation strains.
11. The roles of CsiD, LhgO and CsiR in the stationary phase worth the authors' further examination. For example, this pathway has competitive advantages towards the classic glutaryl-CoA dehydrogenation pathway? How the expression of CsiD and LhgO changes along with CsiR

knockout?

12. The authors proposed that "Lysine degradation via glutarate and L-2-hydroxyglutarate to succinate establishes an important link in central metabolism". To make direct and solid proof, ¹³C labeled experiments to show lysine recycling to central metabolites in special growth phases may be helpful.

Finally, there were so many unacceptable errors in the manuscript. For example, there were two Fig. S5, Fig.S10. Table S1 was absent but Table S2 was present in SI. The sequence of Fig. S1-S4 was also totally wrong. This reviewer kindly suggests the authors thoroughly checking up and modifying the manuscript before submitting this manuscript to other journals.

Reviewer #2:

Remarks to the Author:

The proposed article demonstrates in *E. coli* the existence of a central metabolic pathway for the catabolism of lysine into succinate. Overall, the paper is well written and the methodology is sound. However, a major issue with the proposed manuscript is that the findings described are in highly close proximity with a recent communication from M. Zhang et al. (Nature Communications 9, Article number: 2114 (2018)), published online May 29th 2018, submitted sept. 2017). All key steps of the proposed pathway for lysine degradation, i.e. a glutarate catabolism pathway involving alpha-ketoglutarate dependant stereospecific conversion of glutarate by CsiD into L2HG, followed by conversion of L2HG into alphaKG catalysed by LhgO, and finally CsiD-catalysed conversion into succinate are extensively described in this published article for *Pseudomonas putida* KT2440.

This recent and crucial contribution is not cited in the submitted manuscript (and was perhaps not yet brought to the attention of the authors at the time of submission). This nonetheless strongly challenges the novelty of the proposed article, and its suitability for publication in Nature Communications.

Besides these key findings, a number of interesting results are soundly presented in the proposed contribution, such as the absence of inhibition of CsiD by L2HG, or the detailed characterization of LhgO as a membrane-bound catalyzer of L2HG conversion, and overall embedment of the described pathway in the broader lysine degradation process. These new findings and altogether the results of this interesting study certainly deserve publication of the manuscript in a more specialized journal.

Other comments:

-An independent reference list is inserted at the end of the Methods section p.20. I believe both reference sections should be concatenated into a single list of references.

-The numbering of the first four figures in the supplementary file is incorrect.

Reviewer #3:

Remarks to the Author:

In this manuscript, the authors aim to establish a catabolic pathway of lysine in *E. coli*. While they present strong evidences for the key step (catalyzed by CsiD) of the pathway, most other steps lack sufficient evidences. While in vivo isotopic labeling experiment can be very valuable for establishing the pathway, data for some key metabolites are not shown. Thus, in my opinion, the proposed pathway as shown in Fig. 1 is not established.

Major comments:

1. With NMR analysis (Fig 2a) and genetic experiment (Extended Data Fig. 4), the reaction catalyzed by CsiD converting aKG and glutarate to succinate and L2HG is convincingly

established.

2. Regarding the conversion between aKG and L2HG, while it is demonstrated that LhgO is the catalyzing enzyme, it is not clear whether UQ1 is the major electron acceptor. Fig. 3c only shows that in vitro with UQ1 LhgO shows the highest activities. It does not mean that UQ1 is the major electron acceptor in vivo.
3. There is no data presented to establish the steps from Cad to AVA.
4. There is no direct data to establish the steps from AVA to glutarate. Extended Data Fig. 7a is used by the authors for this purpose. But as far as I can tell from the figure legend, the data shows only the dehydrogenase activity of GabT and GabD, not the production of glutarate.
5. The in vivo isotopic labeling experiment could be a key experiment for establishing the pathway. The authors, however, only show the labeling data for three metabolites, Cad, glutarate, and succinate. It thus remains unclear the path from Cad to glutarate, which also lacks sufficient in vitro data support as I described in points 3 and 4. Can the authors provide the labeling data for the intermediate metabolites? Also, please provide the labeling patterns of the metabolites, which can be useful for distinguishing between alternative pathways. For example, with uniformly ^{13}C labeled lysine, the appearance of $[\text{M}+4]$ succinate would support the proposed pathway by the authors over the canonical pathway through acetyl-CoA.

Minor comments:

1. Extended Data Figs. 1-4 are mislabeled as Extended Data Figs. 6-9, respectively.
2. Some of the bars in Fig. 3c are not labeled.
3. Extended Data Figures with key information should be moved back to the main figures. These include Extended Data Figs. 4 and 7.

Response to Reviewers' comments:

Reviewer #1 (Remarks to the Author):

Review comments for Nature Communications manuscript NCOMMS-18-19566-T, Wide-spread bacterial lysine degradation proceeding via glutarate and L-2-hydroxyglutarate

Major comments:

The authors report a bacterial lysine degradation pathway that proceeds via glutarate and L-2-hydroxyglutarate with CsiD and LhgO as the key enzymes in *Escherichia coli*. Unfortunately, the pathway is essentially the same as that in *Pseudomonas*, which also involves CsiD and LhgO (Nat Commun, 2018, 9:2114). It is likely that the authors did not read the paper since the paper was not cited in the manuscript. In comparison, this manuscript (Knorr et al.) reports the functions of CsiD, LhgO, GabT, GabD and CsiR in *E. coli*. In previous works, the functions of CsiD, GabT and GabD have been identified in *Pseudomonas putida* (Nat Commun, 2018, 9:2114; Appl Environ Microbiol, 2001, 67:5219-24) and the functions of LhgO and CsiR have been identified in *E. coli* (J Bacteriol, 2008, 190:3793-3798; Mol Microbiol, 2004, 51:799-811). The only new information in this manuscript is related to CsiR. Thus, this manuscript significantly overlaps with the previous reports. Thus, the novelty of Knorr et al.'s manuscript is not enough to be published in a distinct journal such as Nature Communications.

Response: *We apologize that we did not recognize the most recent, very interesting study of CsiD and LhgO (Nat Commun 2018) before submitting our manuscript. However, an overlap of our study with the other publications mentioned is in our opinion not correct:*

Appl Environ Microbiol, 2001, 67:5219-24 describes genetic evidence for a pathway from lysine to glutarate in P. putida. We demonstrate for the first time that E. coli possesses dehydrogenases and aminotransferases that are capable of carrying out these reactions. In general, the degradation of lysine to glutarate has been demonstrated in several Pseudomonads and in C. glutamicum, but this is not the key finding of our study. This was already mentioned and cited in the first version of our manuscript.

J Bacteriol, 2008, 190:3793-3798 describes the oxidation of L2HG to aKG and was also cited and discussed in the first version of our manuscript. However, in contrast to the 2008 study - and also in contrast to the 2018 Nat Commun study - we demonstrate that the enzyme is a dehydrogenase rather than an oxidase that couples the reaction to the respiratory chain. See below for further experiments substantiating this finding.

Mol Microbiol, 2004, 51:799-811 describes CsiR as a repressive transcription factor of the csiD operon. The study was also already discussed in the first version. In our study however we demonstrate that glutarate serves as a ligand that inhibits the repressive effect of CsiR.

Specific Comments:

1. This reviewer is fully convinced of the need of in vivo studies providing the proposed roles of the enzymes studied in this work. Specially, the growth of every mutant in M9 medium containing lysine, 5-aminovalerate, glutarate, and L-2-hydroxyglutarate should be assayed.

Response: *We agree that such experiments would be helpful in order to further characterize the pathway. However, please note that E. coli (in contrast to many Pseudomonads) is not able to utilize any of the mentioned compounds as the sole carbon and energy source for heterotrophic growth. In order to address the issue, we have carried out prolonged growth experiments that prove the*

literature-known phenomenon that E. coli is not able to utilize lysine or its degradation products as sole carbon source. We have utilized all compounds as C-sources as suggested by the reviewer (and additionally all N-containing compounds as N-sources), the results are now shown as Extended Data Figure 10. In contrast, we demonstrate for the first time by isotope tracing that after growth in M9 medium with glucose as carbon source, lysine gets degraded via the described pathway. We now have added more data such as the isotope labeling pattern in the detected products as requested by reviewer 3 that fully support our hypotheses, see below for details.

2. In a previous work (J Bacteriol, 2008, 190:3793-3798), LhgO was identified as an oxidase. In this work, the same protein was identified as a ubiquinone-linked dehydrogenase. The authors should do more experiments to support their claims.

Response: *We carefully examined the LhgO-dependent reaction with regard to O₂ consumption utilizing a Clarke O₂-electrode. We did not detect any O₂ consumption with isolated LhgO but the reaction proceeded efficiently with other added electron acceptors such as quinones and DCPIP, as stated in the manuscript. However, when the oxidation of L2HG was carried out with membrane preparations instead of purified LhgO, O₂ consumption was detectable. This consumption was susceptible to a respiratory chain inhibitor, thus, represented the terminal oxidation step. This is just one example of several experiments that we presented in the first version of the manuscript that already contained strong evidence for coupling this reaction to the ETC. This coupling also makes more sense with regard to the membrane localization, energy utilization, and the circumvention of generating the harmful product H₂O₂. In addition, bacteria couple many dehydrogenase reactions to the reduction of quinones in the electron transfer chain. In E. coli alone, 15 different primary dehydrogenases are described to be coupled to the respiratory chain (Unden et al. Biochim Biophys Acta 1320 (1997), 217). We provide convincing evidence that L2HG oxidation further expands this list. However, in order to address the reviewers' request we have done additional experiments that further demonstrate the linkage between LhgO and ubiquinone. We carried out L2HG oxidation reactions catalyzed by E. coli membrane preparations. We then analyzed the ratio of ubiquinol vs. ubiquinone in the membrane. A significant increase of the ubiquinol/ubiquinone ratio was detectable via LC/MS in a L2HG- as well as LhgO-dependent manner. See also responses to reviewer 3 for more details.*

3. Since the regulation mechanism of CsiR is the only new information in the manuscript, the authors should pay more attention on this part. Some important experiments such as EMSA, lacZ reporter fusions should be conducted. The results of footprinting should also be further confirmed by EMSA.

Response: *The EMSA assay is a possibility to characterize the binding between the DNA and csiR. We have already utilized two independent assays (dotblot/filter binding and SPR) that clearly show csiR affinity to a stretch of DNA from the csiD promoter region. Importantly, these assays also allow for characterizing the glutarate-dependency of the binding affinity, as we had already demonstrated in the manuscript. With regard to confirming the footprint with an EMSA assay: The footprint was carried out in order to precisely identify the binding sites in the promoter region. The suggestion to utilize LacZ fusions in order to demonstrate the regulatory potential of csiR is certainly a helpful comment. There is already a very careful analysis of the regulation of the csiD operon carried out by Hengge and co-workers (Mol Microbiol, 281 51, 799-811 (2004)) that was already mentioned in the original submission. The publication contains a series of lacZ fusion experiments that clearly demonstrate that csiR suppresses transcription of the csiD operon. What we add to this information however is the finding that glutarate is the ligand that in turn controls csiR. We have now added a sentence to the paragraph describing csiR-based control of the csiD operon that explicitly states the findings by Hengge and co-workers.*

4. In abstract, what is the meaning of “dead-end metabolite”? Glutarate can be metabolized to acetyl-CoA through a series of reactions. It is not suitable to define glutarate as a “dead-end metabolite”.

Response: *We did not coin this expression: Since in humans glutarate is not an intermediate but glutaryl-CoA, and there are no other known ways of generating and/or degrading glutarate except via glutaryl-CoA, it has been termed “dead-end-metabolite”. However, in light of the new findings it is certainly no more a dead-end-metabolite (at least not in bacteria), which is what we wanted to express. Anyhow, we have removed the expression in accordance with the reviewer’s suggestion.*

5. In abstract section the authors introduced possible relation between gut microbiome and the chemicals studied in this manuscript. However, there was no detailed discussion about this issue in the main text. No data was involved to support the view in the abstract “Our work introduces the gut microbiome as a potential source of these metabolites associated with human diseases such as cancer and organic acidurias”.

Response: *This issue refers to the last sentence of the abstract that should serve to put the findings into a broader context. We have discussed this issue in the last paragraph of the main text. We are convinced that it should be possible to mention potential implications in the discussion. From our point of view, it would be very interesting to investigate the gut microbiome as a potential source of glutarate and hydroxyglutarates but such a study is certainly beyond the scope of the present work. We have rephrased the sentence in the abstract to “Our results may imply the gut microbiome as a potential source of these metabolites associated with human diseases such as cancer and organic acidurias”*

6. The abstract, Figure 1 and the legend should be revised to highlight the regulation mechanism of CsiR instead of the identification of the pathway.

7. Correspondingly, extended Data Fig. 8, Fig. 9 and Fig. 10 might be better to locate in the main text to characterize CsiR.

8. Site-specific mutagenesis is required to determine the key sites involved in the regulation process of CsiR.

Response to issues 6-8: *We would like to refrain from re-focusing the whole work on CsiR for the reasons mentioned above.*

9. To demonstrate “lhgo is the only L-2-hydroxyglutarate-converting enzyme in E. coli”, on the basis of the results of enzyme assay of Δ lhgo, the author might add some growth experiments with lysine or glutarate as the sole carbon source and thus determine the intracellular change of L-2-hydroxyglutarate.

Response: *Please see response to issue 1 above, these experiments are not possible for this reason. In order to address this issue, we have removed the statement from the manuscript text.*

10. Some key experiments based on the knockout strains are better to conduct again with gene-complementation strains.

Response: We have carried out a complementation experiment where membrane-bound LhgO is oxidizing added L2HG. This activity is lost in the *lhgO* knockout and we reconstituted the activity by overexpressing a complemented *lhgO* construct. The data is shown in Figure 3a.

11. The roles of CsiD, LhgO and CsiR in the stationary phase worth the authors' further examination. For example, this pathway has competitive advantages towards the classic glutaryl-CoA dehydrogenation pathway? How the expression of CsiD and LhgO changes along with CsiR knockout?

Response: *E. coli* does not possess the mentioned glutaryl-CoA dehydrogenation pathway, hence a comparison is not possible in a way as it was nicely demonstrated in *P. putida* in the recent *Nat Commun* 2018 publication. With regard to the CsiR knockout influencing CsiD and LhgO levels: Upon deletion of *csiR*, the expression of both enzymes is induced, as was already described in detail in *Mol Microbiol*, 281 51, 799-811 (2004). We have discussed this finding in accordance with the reviewer's suggestion, see also response to issue 3 above.

12. The authors proposed that "Lysine degradation via glutarate and L-2-hydroxyglutarate to succinate establishes an important link in central metabolism". To make direct and solid proof, ¹³C labeled experiments to show lysine recycling to central metabolites in special growth phases may be helpful.

Response: The first version already described these experiments in detail, demonstrating the degradation of lysine to succinate via ¹³C-labeling and LC/MS. We have now added further details regarding the detection of intermediates and the inclusion of isotope patterns that proves the mentioned link to central metabolism. For details see Response to Reviewer 3, Issue 5.

Finally, there were so many unacceptable errors in the manuscript. For example, there were two Fig. S5, Fig.S10. Table S1 was absent but Table S2 was present in SI. The sequence of Fig. S1-S4 was also totally wrong. This reviewer kindly suggests the authors thoroughly checking up and modifying the manuscript before submitting this manuscript to other journals.

Response: Thank you. These errors are corrected in our revised manuscript.

Reviewer #2 (Remarks to the Author):

The proposed article demonstrates in *E. coli* the existence of a central metabolic pathway for the catabolism of lysine into succinate. Overall, the paper is well written and the methodology is sound.

However, a major issue with the proposed manuscript is that the findings described are in highly close proximity with a recent communication from M. Zhang et al. (*Nature Communications* 9, Article number: 2114 (2018)), published online May 29th 2018, submitted sept. 2017). All key steps of the proposed pathway for lysine degradation, i.e. a glutarate catabolism pathway involving alpha-ketoglutarate dependant stereospecific conversion of glutarate by CsiD into L2HG, followed by conversion of L2HG into alphaKG catalysed by LhgO, and finally CsiD-catalysed conversion into succinate are extensively described in this published article for *Pseudomonas putida* KT2440. This recent and crucial contribution is not cited in the submitted manuscript (and was perhaps not yet brought to the attention of the authors at the time of submission). This nonetheless strongly challenges the novelty of the proposed article, and its suitability for publication in *Nature Communications*.

Besides these key findings, a number of interesting results are soundly presented in the proposed contribution, such as the absence of inhibition of CsiD by L2HG, or the detailed characterization of LghO as a membrane-bound catalyzer of L2HG conversion, and overall embedment of the described pathway in the broader lysine degradation process. These new findings and altogether the results of this interesting study certainly deserve publication of the manuscript in a more specialized journal.

Response: *We thank this reviewer for the encouraging comments. Taking into account the editorial suggestion of overruling the lack of novelty and with the additional data such as the crystal structures added to the revised manuscript, we hope that our work is now acceptable for publication in Nature Communications.*

Other comments:

- An independent reference list is inserted at the end of the Methods section p.20. I believe both reference sections should be concatenated into a single list of references.
- The numbering of the first four figures in the supplementary file is incorrect.

Response: *We have merged the references and corrected the order of the figures in the SI file.*

Reviewer #3 (Remarks to the Author):

In this manuscript, the authors aim to establish a catabolic pathway of lysine in *E. coli*. While they present strong evidences for the key step (catalyzed by CsiD) of the pathway, most other steps lack sufficient evidences. While in vivo isotopic labeling experiment can be very valuable for establishing the pathway, data for some key metabolites are not shown. Thus, in my opinion, the proposed pathway as shown in Fig. 1 is not established.

Response: *In the initial version we concentrated on the key findings of CsiD and LghO, since the upper pathway of lysine to aminovalerate degradation is already known in some Pseudomonads and homologs of the involved enzymes are present in E. coli. However, as requested, we have now characterized the E. coli enzymes in more detail. We also added further experiments regarding the LghO reaction and substantiated the isotope labeling reactions as suggested, see below for details.*

Major comments:

1. With NMR analysis (Fig 2a) and genetic experiment (Extended Data Fig. 4), the reaction catalyzed by CsiD converting aKG and glutarate to succinate and L2HG is convincingly established.
2. Regarding the conversion between aKG and L2HG, while it is demonstrated that LghO is the catalyzing enzyme, it is not clear whether UQ1 is the major electron acceptor. Fig. 3c only shows that in vitro with UQ1 LghO shows the highest activities. It does not mean that UQ1 is the major electron acceptor in vivo.

Response: *As already mentioned in response to reviewer 1, issue 2, we have now achieved a direct prove of reduction of quinones by LghO utilizing membrane preparations. Addition of L2HG increases the ratio of ubiquinole/ubiquinone. This activity is not observed in the Δ lghO strain. We also analysed the two naphtoquinones present in E. coli. We did not detect reduction upon addition of L2HG, however the naphtoquinones are more sensitive regarding re-oxidation and we cannot completely rule out that naphtoquinones may be electron acceptors as well. However, ubiquinone is the main e-acceptor under aerobic conditions in E. coli, whereas the naphtoquinones are primarily utilized in anaerobic respiration. Hence, our observations regarding quinone reduction fit well to the O₂-*

requirement of the CsiD reaction. We have critically discussed this issue in detail in the revised version.

3. There is no data presented to establish the steps from Cad to AVA.

Response: We have now purified PatA and PatD from *E. coli* and demonstrate that the conversion from Cad to AVA is catalyzed by these enzymes efficiently. These new experiments are now included in Figure 4 and Extended Data Figure 9 and discussed in the main text.

4. There is no direct data to establish the steps from AVA to glutarate. Extended Data Fig. 7a is used by the authors for this purpose. But as far as I can tell from the figure legend, the data shows only the dehydrogenase activity of GabT and GabD, not the production of glutarate.

Response: We have now demonstrated the production of glutarate by analysis of the retention time and HRMS of the products of the respective enzyme reaction in comparison to glutarate as standard. This new result is mentioned in the main text and shown in the Extended data file.

5. The in vivo isotopic labeling experiment could be a key experiment for establishing the pathway. The authors, however, only show the labeling data for three metabolites, Cad, glutarate, and succinate. It thus remains unclear the path from Cad to glutarate, which also lacks sufficient in vitro data support as I described in points 3 and 4. Can the authors provide the labeling data for the intermediate metabolites? Also, please provide the labeling patterns of the metabolites, which can be useful for distinguishing between alternative pathways. For example, with uniformly ¹³C labeled lysine, the appearance of [M+4]succinate would support the proposed pathway by the authors over the canonical pathway through acetyl-CoA.

Response: We have now additionally confirmed and quantified the intermediates aminopentanal and aminovalerate. Aminovalerate however could only be quantified in its labelled form since its detection is hampered by valine of similar retention time and the same sum formula. Further, we were not able to detect succinate semialdehyde, likely because of the instable nature of this compound. L-2-HG however was only detectable in low amounts, but was difficult to quantify. These new results are now included in Table S2. We have also analysed the labeling pattern as requested, and we are grateful for this suggestion, since it further validates the pathway. This information is also included in Table S2. For the immediate degradation products, predominantly fully labeled intermediates are detected. However, for succinate, predominantly unlabeled as well as M+2 and M+4 masses are detectable. This finding is in accordance with other sources for this central metabolite and the turnover of succinate via the TCA when originating from degradation of fully labeled lysine. These findings are now discussed in detail in the main text.

Minor comments:

1. Extended Data Figs. 1-4 are mislabeled as Extended Data Figs. 6-9, respectively.

Response: Thank you. We corrected the labeling of the Figures.

2. Some of the bars in Fig. 3c are not labeled.

Response: We have clarified the labeling of the respective data.

3. Extended Data Figures with key information should be moved back to the main figures. These include Extended Data Figs. 4 and 7.

Response: We have moved the respective content to the main manuscript as requested, now depicted in Figure 2d and Figure 4.

Reviewers' Comments:

Reviewer #1:

Remarks to the Author:

The authors added new data in the revised manuscript. Now the most important new information in this manuscript is the structure of CsiD. Several issues (concerns) should be addressed.

As for CsiD:

1, The authors declared CsiD is not inhibited by L-2-hydroxyglutarate. However, it is inhibited by L-2-hydroxyglutarate as shown in Extended Data Fig. 5. The authors should add new data to support this opinion. On the other hand, the K_i of L-2-hydroxyglutarate, D-2-hydroxyglutarate and NOG on CsiD should be calculated based on more data. Reactions at a specific concentration of L-2-hydroxyglutarate, D-2-hydroxyglutarate and NOG can not get the K_i of these compounds.
2, CsiD is able to discriminate 2-KG and succinate could not reveal why CsiD is not inhibited by L-2-hydroxyglutarate. It is the discrimination between 2-KG and L-2-hydroxyglutarate to support the opinion. The more important structure of CsiD with L-2-hydroxyglutarate should be added.

As for LhgO:

1, The purification process and the results should be described.
2, The basic biochemical characterization of this enzyme is needed. The authors should consider providing this information. At least, k_m for each substrate, V_{max} , k_{cat} etc should be provided.

Other specific comments:

1, this reviewer is obsessed by the function of gabP described in Fig. 1. If this protein can transport 5-AVA, why *E. coli* can not utilize 5-AVA as described? The authors should consider providing more data to reveal this question.
2, the authors should add some discussion related to the reaction mechanism of CsiD. More mutation data are welcome to give the readers more understanding about the enzyme.
3, the authors should provide a table describing the kinetic parameters of the enzymes introduced in this work.

Reviewer #3:

Remarks to the Author:

With the additional data, the authors have now convincingly established the proposed pathway in Fig. 1.

Reviewer #4:

Remarks to the Author:

In the manuscript entitled "Wide-spread bacterial lysine degradation proceeding via glutarate and L-2-hydroxyglutarate" by Knorr et al. (Manuscript NCOMMS-18-19556-A) the authors investigate the lysine degradation in *E. coli*.

To the reviewer's point of view the current version of the manuscript is very well written and comprehensive. In the revised version of the manuscript the authors have included several crystal structures of the CsiD protein: CsiD Apo, CsiD with bound glutarate, CsiD with bound N-oxalylglycine and CsiD with bound succinate. This structural work is very supportive for the author's discoveries. Moreover this is a clear supplement compared to the very recent publication by Zhang et al. *Nat Commun*, 2018, 9:2114.

The here presented structures are of sufficient resolution to justify the conclusions drawn by the authors. Interestingly the authors revealed two binding sites for the ligands termed "substrate site" and "co-substrate site". The conformation of the protein seems to be unaffected upon ligand binding.

Minor points:

- Line 124, remove "fundamental". It is enough to state that the two copies are identical
- Figure 2e: Explain in the figure legend the "dashed lines"
- Line 334. "*P. putida*" in italic
- Please acknowledge the beamline access to the SLS
- Frequently spaces are missing between value and unit within the Materials&Methods section describing the crystallographic experiments
- Line 462: correct the sentence to "... Crystallization was performed in 96-well plates...."
- Line 474: rephrase the sentences to "Phasing was performed by molecular"
- Line 478: correct the sentence to "... manual model building was performed in COOT."
- Provide a references for PDB-ID 2R6S
- Extended Data Fig.6: Please use same scale for the structures, panel (a) and (b). If possible use identical views on the active site. Currently panel (a) and (b) have an identical view, as well as panel (c) and (d). Why the authors show a 2FoFc electron density map for the iron?
- Maybe provide for the ligand structures an additional figure with an omit electron density with the respective ligand omitted.

Bernhard Loll

Point-by-point response to reviewers' comments

REVIEWERS' COMMENTS:

Reviewer #1 (Remarks to the Author):

The authors added new data in the revised manuscript. Now the most important new information in this manuscript is the structure of CsiD. Several issues (concerns) should be addressed.

As for CsiD:

1, The authors declared CsiD is not inhibited by L-2-hydroxyglutarate. However, it is inhibited by L-2-hydroxyglutarate as shown in Extended Data Fig. 5. The authors should add new data to support this opinion. On the other hand, the K_i of L-2-hydroxyglutarate, D-2-hydroxyglutarate and NOG on CsiD should be calculated based on more data. Reactions at a specific concentration of L-2-hydroxyglutarate, D-2-hydroxyglutarate and NOG can not get the K_i of these compounds.

Response: We changed the statement in Extended Data Fig. 5 (Supplementary Figure 5). 5 mM of L2HG is not sufficient to reduce the activity of CsiD to its half-maximum, thus it is reasonable to estimate the K_i to be higher than 5 mM. Therefore, the K_i for CsiD is at least 10-fold higher than reported for other α KG dependent dioxygenases. Furthermore, the K_i of L2HG was reported to be equal to that of NOG (Xu et al., 2011), which is clearly not the case in our experiments. In accordance to Xu et al., reporting D2HG as a weak inhibitor for α KG dependent dioxygenases, although the K_i is above 10 mM, we changed our statement in the main text to L2HG being a weak inhibitor for CsiD.

2, CsiD is able to discriminate 2-KG and succinate could not reveal why CsiD is not inhibited by L-2-hydroxyglutarate. It is the discrimination between 2-KG and L-2-hydroxyglutarate to support the opinion. The more important structure of CsiD with L-2-hydroxyglutarate should be added.

Response: Significant effort was devoted to obtaining this structure using both crystal soaking and co-crystallization methods. However, crystals of appropriate diffraction quality could not be obtained for this complex.

As for LhgO:

1, The purification process and the results should be described.

Response; The purification for LhgO was conducted as stated in section "Protein purification". We added more details regarding the process and buffer exchange of LhgO to the methods section, "Activity of purified LhgO and membrane fractions".

2, The basic biochemical characterization of this enzyme is needed. The authors should consider providing this information. At least, k_m for each substrate, V_{max} , k_{cat} etc should be provided.

Response: A thorough characterization of E. coli LhgO was already performed by Kalliri et al. cited at the respective position in the manuscript.

Other specific comments:

1, this reviewer is obsessed by the function of gabP described in Fig. 1. If this protein can transport 5-AVA, why E. coli can not utilize 5-AVA as described? The authors should consider providing more data to reveal this question.

Response: E. coli is able to utilize 5-AVA as stated by our labeling experiments where a gabT knockout accumulates 5-AVA (Suppl. Table 3). Furthermore, a gabP knockout shows impaired growth on 5-AVA as sole nitrogen source indicating the effect of 5-AVA transport by gabP (Suppl. Fig. 10). The inability of E. coli to utilize 5-AVA (and all other compounds involved in the pathway) as sole carbon source is not surprising. The presence of a transporter or even a catabolic pathway does not necessarily mean that a bacterium is able to grow on a single metabolite present in an artificial medium. For example the close E. coli relative Klebsiella aerogenes is able to grow on

arginine and its breakdown products putrescine or GABA as sole carbon-and nitrogen-source (Friedrich, B. & Magasanik, B. Enzymes of agmatine degradation and the control of their synthesis in Klebsiella aerogenes. Journal of bacteriology 137, 1127-1133 (1979)), whereas E. coli is only able to use them as nitrogen-source (Reitzer, L. & Schneider, B. L. Metabolic context and possible physiological themes of sigma(54)-dependent genes in Escherichia coli. Microbiology and molecular biology reviews : MMBR 65, 422-444, table of contents, doi:10.1128/membr.65.3.422-444.2001 (2001)). In case of arginine the difference is caused by a stronger and more proper regulated promotor of the ast operon (Schneider, B. L., Kiupakis, A. K. & Reitzer, L. J. Arginine catabolism and the arginine succinyltransferase pathway in Escherichia coli. Journal of bacteriology 180, 4278-4286 (1998)). Differences in regulation and transport might also be the reason why E. coli is unable to grow on lysine and its breakdown products. However, our results support that lysine degradation via the csiD-operon contributes to adaption for stationary phase conditions as seen in our growth experiment with a csiD knockout mutant (Fig. 2d & Suppl. Fig. 15).

2, the authors should add some discussion related to the reaction mechanism of CsiD. More mutation data are welcome to give the readers more understanding about the enzyme.

Response: CsiD displays features (both kinetically as well as structurally) that are in full agreement with the general reaction mechanism of aKG-dependent dioxygenases. In general, the exact reaction mechanism is still under debate and an active field of research. The requested experiments are beyond the scope of the present study.

3, the authors should provide a table describing the kinetic parameters of the enzymes introduced in this work.

Response: The kinetic parameters for the different enzymes are given at the respective positions in the main text and figures. We do not agree that summing up the functionally diverse enzymes characterized in the study in one table would improve the representation of the results.

Reviewer #3 (Remarks to the Author):

With the additional data, the authors have now convincingly established the proposed pathway in Fig. 1.

Reviewer #4 (Remarks to the Author):

In the manuscript entitled "Wide-spread bacterial lysine degradation proceeding via glutarate and L-2-hydroxyglutarate" by Knorr et al. (Manuscript NCOMMS-18-19556-A) the authors investigate the lysine degradation in E. coli.

To the reviewer's point of view the current version of the manuscript is very well written and comprehensive. In the revised version of the manuscript the authors have included several crystal structures of the CsiD protein: CsiD Apo, CsiD with bound glutarate, CsiD with bound N-oxalylglycine and CsiD with bound succinate. This structural work is very supportive for the author's discoveries. Moreover this is a clear supplement compared to the very recent publication by Zhang et al. Nat Commun, 2018, 9:2114.

The here presented structures are of sufficient resolution to justify the conclusions drawn by the authors. Interestingly the authors revealed two binding sites for the ligands termed "substrate site" and "co-substrate site". The conformation of the protein seems to be unaffected upon ligand binding.

Minor points:

- Line 124, remove "fundamental". It is enough to state that the two copies are identical

Done

- Figure 2e: Explain in the figure legend the “dashed lines”

Done

- Line 334. “P. putida” in italic

Done

- Please acknowledge the beamline access to the SLS

Done

- Frequently spaces are missing between value and unit within the Materials&Methods section describing the crystallographic experiments

Done

- Line 462: correct the sentence to “... Crystallization was performed in 96-well plates.....”

Done

- Line 474: rephrase the sentences to “Phasing was performed by molecular

Done

- Line 478: correct the sentence to “... manual model building was performed in COOT.”

Done

- Provide a references for PDB-ID 2R6S

Hyperlink and citation inserted was inserted (line 394)

- Extended Data Fig.6: Please use same scale for the structures, panel (a) and (b). If possible use identical views on the active site. Currently panel (a) and (b) have an identical view, as well as panel (c) and (d).

Response: The scale of the panels is adjusted to the size of the object they display. Panel a only shows the groups coordinating the iron ion as this panel corresponds to the apo-active site void of any bound compounds. Panels b, c and d show the active site complexed to a ligand and necessarily must display a larger portion of space than panel a. In other words, panel a displays a subset of the space included in the other panels. Down-scaling panel a to fit the size of the same atoms in panels b,c and d would result in a reduction of the object displayed, which would then stand surrounded by a large, empty white space. This would reduce the visibility of the object without bringing any advantage to the figure. Under this consideration, we prefer to keep the current scale.

Regarding the orientation of the models in the panels, this has been chosen as to offer optimal visibility of the ligands shown. The crystal structure is a 3D object and a simultaneous clear visualization of the substrate and co-substrate sites in the 2D plane of the paper is not achievable when electron density maps must be displayed. Panel c and d have a same orientation as to allow the direct visual comparison of compounds NOG and SA that occupy the same site. However, panel b necessarily displays a different orientation as to permit visualizing the GA binding site instead.

- Why the authors show a 2FoFc electron density map for the iron?

Response: We included the electron density at this locus in order to demonstrate the heavy nature of the atom in support of the expectation of iron binding at that position, which is otherwise well established in this protein family.

- Maybe provide for the ligand structures an additional figure with an omit electron density with the respective ligand omitted. Bernhard Loll

Response: The requested omit density maps are now provided in a revised Supp. Fig. 6.